# Diversity, Taxonomic Novelty, and Encoded Functions of Salar de Ascotán Microbiota, as Revealed by Metagenome-Assembled Genomes

**DOI:** 10.3390/microorganisms11112819

**Published:** 2023-11-20

**Authors:** Marcelo Veloso, Angie Waldisperg, Patricio Arros, Camilo Berríos-Pastén, Joaquín Acosta, Hazajem Colque, Macarena A. Varas, Miguel L. Allende, Luis H. Orellana, Andrés E. Marcoleta

**Affiliations:** 1Grupo de Microbiología Integrativa, Laboratorio de Biología Estructural y Molecular BEM, Faculty of Science, Universidad de Chile, Las Palmeras 3425, Ñuñoa, Santiago 7800003, Chile; marcelo.veloso@ug.uchile.cl (M.V.); angie.waldisperg@ug.uchile.cl (A.W.); patricio.arros@ug.uchile.cl (P.A.); camilo.berrios.p@ug.uchile.cl (C.B.-P.); joaquin.acosta@ug.uchile.cl (J.A.); hazajem.colque@ug.uchile.cl (H.C.); macarena.varas@uchile.cl (M.A.V.); 2Millenium Institute Center for Genome Regulation, Faculty of Science, Universidad de Chile, Las Palmeras 3425, Ñuñoa, Santiago 7800003, Chile; mallende@uchile.cl; 3Department of Molecular Ecology, Max Planck Institute for Marine Microbiology, Celsiusstr. 1, D-28359 Bremen, Germany; lorellan@mpi-bremen.de

**Keywords:** salt flat, extremophiles, novel archaea, resistance, hybrid metagenomic assembly

## Abstract

Salar de Ascotán is a high-altitude arsenic-rich salt flat exposed to high ultraviolet radiation in the Atacama Desert, Chile. It hosts unique endemic flora and fauna and is an essential habitat for migratory birds, making it an important site for conservation and protection. However, there is limited information on the resident microbiota’s diversity, genomic features, metabolic potential, and molecular mechanisms that enable it to thrive in this extreme environment. We used long- and short-read metagenomics to investigate the microbial communities in Ascotán’s water, sediment, and soil. Bacteria predominated, mainly *Pseudomonadota*, *Acidobacteriota*, and *Bacteroidota*, with a remarkable diversity of archaea in the soil. Following hybrid assembly, we recovered high-quality bacterial (101) and archaeal (6) metagenome-assembled genomes (MAGs), including representatives of two putative novel families of *Patescibacteria* and *Pseudomonadota* and two novel orders from the archaeal classes *Halobacteriota* and *Thermoplasmata*. We found different metabolic capabilities across distinct lineages and a widespread presence of genes related to stress response, DNA repair, and resistance to arsenic and other metals. These results highlight the remarkable diversity and taxonomic novelty of the Salar de Ascotán microbiota and its rich functional repertoire, making it able to resist different harsh conditions. The highly complete MAGs described here could serve future studies and bioprospection efforts focused on salt flat extremophiles, and contribute to enriching databases with microbial genome data from underrepresented regions of our planet.

## 1. Introduction

Among the natural extreme environments found on our planet, the salt flats in the South American Altiplano stand out given their exceptional climatic conditions, influenced by their considerable altitude and geographic location. These factors determine significant thermal oscillations, low atmospheric pressure, reduced humidity levels, and low atmospheric oxygen, giving rise to a semi-arid climate with the impact of high solar UV radiation [1,2]. Thus, the biological niches in these areas are valuable for studying biodiversity and adaptation strategies, hosting yet-to-be-explored microbial communities and biotechnological potential.

One of these environments corresponds to Salar de Ascotán, located north of the Chilean Altiplano in the Atacama Desert, at an average altitude of 3724 m above sea level (m.a.s.l.). It encompasses a 2.43 km^2^ salt crust and a total area of 18 km^2^, including lagoons and springs, and is surrounded by a volcanic chain [3]. Unlike talasohaline environments, which have a marine origin and saline composition dominated mainly by sodium chloride, Salar de Ascotán is atalasohaline, and rich in other ions, including chlorides, sulfates, and boron. Additionally, arsenic sulfides have been detected in the highest concentrations in the area, representing an elevated toxicity risk for most organisms [4,5]. Moreover, extreme thermal variations due to its geographic conditions, including intense solar exposure and strong winds, contribute significantly to the evaporation processes that characterize this environment [6].

Even under the harsh environmental conditions mentioned above, Salar de Ascotán and other salt flats in the Chilean Altiplano cover large areas with important ecosystems hosting different life forms [6,7]. Among them are at least 21 plant species, predominately hemicryptophytes in water-covered areas and nanophanerophytes shrubs in the surrounding slopes [8], and the small ray-finned fish *Orestias ascotanensis* (Cyprinodontidae), which is endemic to this salt flat. Furthermore, given the isolation of this environment, at least since the elevation of the Andes in the late Miocene (11.6 to 5.3 million years ago), this zone was proposed as a natural laboratory to study aquatic populations’ speciation and diversification [9]. In this line, genomic and transcriptomic analyses have revealed the expansion of genes that preserve genome stability as a possible adaptation of *O. ascotanensis* to high UV radiation and altitude [10]. Despite this valuable information, there is still limited knowledge regarding the microbiota inhabiting this environment, especially their genetic properties and metabolic capabilities. Moreover, the extreme conditions of Salar de Ascotán make it an outstanding model environment to study microbial adaptations to extreme salinity, UV radiation, and high altitudes.

Previous studies investigating microbial communities in high-altitude salt flats and wetlands in northern Chile have revealed the presence of a high diversity of halophilic and non-halophilic prokaryotes, with a predominance of bacteria compared to archaea and highly a variable microbial community composition among different sites [6,7]. However, no studies have evaluated the Salar de Ascotán microbiota at a genomic level, meaning there is a lack of information on its encoded metabolic capabilities, including tolerance mechanisms to harsh environmental conditions, and preventing their accurate taxonomic classification. Thus, further investigations are required to understand better the structure and functional processes governing the Ascotán salt flat microbial communities. Moreover, examining the composition of the archaeal lineages is particularly interesting, considering their high representation reported in other salt flats [5,6,7].

Advances in massive DNA sequencing and dedicated bioinformatics analyses have revolutionized the study of microorganisms in the environment, especially in remote and extreme ecosystems [11,12]. DNA-based culture-independent approaches have emerged as a cornerstone, allowing researchers to explore the vast diversity of microorganisms that are notoriously difficult to culture in the laboratory [13]. Metabarcoding, a popular approach, targets specific genetic markers like the 16S rRNA gene or internal-transcribed spacer ITS regions, offering insights into community composition. On the other hand, shotgun sequencing involves the random sequencing of DNA fragments from the total metagenomic DNA sample, enabling researchers to reconstruct genomes and gain functional insights [14]. Meanwhile, short-read sequencing, utilizing technologies like Illumina, provides high-quality (low-error) data, while long-read sequencing such as Oxford Nanopore, although considerably more error-prone, offers lengthier DNA fragments, contributing to improved genome assemblies and the resolution of complex genomic regions. Combining short and long-read data enables the generation of hybrid metagenomic assemblies, significantly increasing the assembled contigs’ length and accuracy. This combination further benefits downstream processes like contigs binning and bin refinement for reconstructing high-quality MAGs from complex datasets [15,16]. These advancements have empowered researchers to unravel environmental microbial communities’ structure, function, and dynamics, yielding insights into their roles in diverse ecosystems and their potential applications across various fields.

In this study, we used 16S rRNA metabarcoding to investigate the diversity and structure of microbial communities inhabiting the Salar de Ascotán. Moreover, we combined Illumina and Nanopore shotgun metagenomic sequencing to recover numerous high-quality MAGs, representatives of the main taxa identified. Additionally, we investigated the metabolic functions encoded in these MAGs and the presence of genes potentially involved in toxic metals and oxidative stress tolerance, gaining insights into the molecular adaptations that the resident microbiota have evolved to thrive in this extreme environment.

## 2. Materials and Methods

### 2.1. Site of Study and Sample Collection

The samples were collected on 9–11 October 2021, in Salar de Ascotán (Atacama Desert, Chilean Altiplano; altitude: ~3300 m.a.s.l.), specifically in stream six and its surroundings (Appendix A). Temperature, pH, and other parameters were measured on site (Table 1). The water samples were collected from a zone close to the pond’s edge, using two 5 L sterile plastic bottles previously washed with water from the pond, as reported previously [7,17]. The bottles were kept at room temperature during the transport to the laboratory. Afterward, they were stored at 4 °C until filtering. A total of 30 L of water was collected during three sampling days. The collected water was filtered using 47 mm in diameter filters with 0.45 µm pores (MF-MiliporeTM MCE Membrane) [18] mounted on a Nalgene^®^ filtration unit connected to a vacuum pump. The filters were removed with sterile tweezers, placed on a sterile Whirl-pak^®^ sampling bag (Nasco, WI, USA), kept at −20 °C during the expedition, and then kept at −80 °C once in the laboratory. Sediment samples were collected in sterile Whirl-pak^®^ bags from the upper 2–3 cm of a low-depth area, using a sterile shovel and removing the excess water, as previously reported [19,20]. The bags were kept at 4 °C until processing. Soil samples were collected following a similar strategy used for surface dry soil close to stream six.

### 2.2. DNA Purification and Quality Assessment

DNA was extracted from sediment and soil samples using the PowerSoil^®^ DNA Isolation Kit (Qiagen, Hilden, Germany), following the manufacturer’s recommendations. DNA extraction from water was performed using the Cells and Tissue DNA Isolation Kit (Norgen Biotek©, Thorold, ON, USA). For this, the filters were cut in halves, and each half was cut into smaller pieces and introduced on a 1.5 mL Eppendorf tube. The scissors were thoroughly washed and then sterilized with ethanol before filter cutting. Then, 200 µL of PBS was added to each tube and mixed via vortexing for 5 s. Following, 20 µL of proteinase K and 300 µL of Lysis buffer B were added, followed by vortex mixing and incubation for 20 min at 55 °C. Next, 10 µL of RNAse A was added, mixed via vortex and incubated at 37 °C for 15 min. Then, 110 µL of absolute ethanol was added and mixed via vortexing. The following steps were performed according to the manufacturer’s recommendations. The final elution was performed with 50 µL of nanopure water. The extracted DNA was quantified using a Qubit fluorometer and the dsDNA High-Sensitivity Assay Kit (ThermoFisher, Waltham, MA, USA), and quality-checked using 1% TAE-agarose gel electrophoresis. Due to the low DNA yield, probably owing to the high salinity of the samples, several DNA extractions from different bags (soil and sediment) and filters (water) were pooled and then used for 16S rRNA metabarcoding and shotgun sequencing (Nanopore and Illumina).

### 2.3. 16S rRNA Metabarcoding and Microbial Diversity Analyses

16S rRNA gene amplicon libraries were constructed and sequenced by hiring the services of Macrogen Inc. (Seoul, Korea). For quality control, soil DNA was used along with the Herculase II Fusion DNA Polymerase Nextera XT Index Kit V2 (Illumina, San Diego, CA, USA) to amplify the V3-V4 region of the 16S rRNA gene using the primers Bakt_341F: CCTACGGGNGGCWGCAG and Bakt_805R: GACTACHVGGGTATCTAATCC. For the directed analysis of archaeal amplicon sequence variants (ASVs), we used the primers Arc_787F: ATTAGATACCCSBGTAGTCC and Arc_1059R: GCCATGCACCWCCTCT. The libraries were sequenced using an Illumina Miseq sequencer, generating ~106,000–186,000 300-bp paired-end reads (~32–56 Mbp) per sample. Microbial diversity analyses, including relative abundance and alpha- and beta-diversity calculations, were performed using Dada2 and the QIIME2 platform v. 2019.7.0 [21,22].

### 2.4. Illumina and Nanopore Shotgun DNA Sequencing

Illumina shotgun metagenomic libraries were prepared using the Truseq DNA PCR Free library Kit (350 bp). Sequencing was performed in an Illumina Novaseq machine, obtaining 150-bp paired-end reads. Nanopore libraries were prepared using 1 µg of soil DNA and the Ligation Sequencing kit LSK-110, following the manufacturer’s guidelines. For each library, 5–50 fmol was loaded into an FLO-MIN106D flow cell and sequenced using a MinION device (Nanopore Technologies Ltd., Oxford, UK). Quality control of the Illumina and Nanopore reads for each metagenome was performed using FastQC v0.11.5 (https://www.bioinformatics.babraham.ac.uk/projects/fastqc/; accessed on 14 November 2022) and NanoPlot v1.33.1 [23], respectively. Nanopore reads were trimmed with NanoFilt v2.7.1 [23] using a minimum threshold of 1000 bp and a Phred Score 7.

### 2.5. Shotgun Sequence Diversity Analyses

Metagenomic distance estimation was performed with Mash [24], inferring a distance tree using Mashtree [25]. Also, the coverage of each metagenome and the captured diversity were evaluated using Nonpareil v3.401 [26]. The taxonomic assignment of the Illumina reads was performed using Kaiju v1.9.0 [27] using the nr_euk (24 February 2021) database, while Kraken v2.1.2 was used for the Nanopore reads [28].

### 2.6. Hybrid Metagenome Assembly and MAGs Reconstruction, Annotation, and Functional Profiling

For hybrid metagenome assembly, Nanopore reads were trimmed and assembled using metaFlye v2.9-b1778 [29], and then the assembly was polished with Medaka v1.7.1 (2018- Oxford Nanopore Technologies Ltd., Oxford, UK). Subsequently, the Illumina reads were filtered using Fastp v0.22.0 [30] and used to polish the long-read assembly using Polypolish v0.5.0 [31]. Finally, the obtained assembly was polished using POLCA from the MaSuRCA v4.0.9 package [32] and evaluated using QUAST v5.0.2 [33].

For MAGs recovery, the hybrid assemblies were used as input for the metaWRAP v1.3.2 pipeline [34], running a combination of the MetaBAT2 [35], CONCOCT [36], and MaxBin2 [37] tools included in the binning module. The obtained bins meeting a minimum quality threshold of 50 (quality = completeness − 5 × contamination) were refined using the metaWRAP bin_refinement module. The taxonomic assignment of the recovered MAGs was performed using GTDB-Tk v2.1.1 [38] using the R207_v2 database. MAG general annotation was performed using Prokka, while the functional and metabolic profiling was performed using DRAM [39] and SUPER-FOCUS [40]. Metal and biocide resistance genes were predicted using Diamond blastx [41] and the BacMet database of experimentally demonstrated metal resistance genes [42].

### 2.7. Phylogenetic Analyses

Phylogenetic inference among the MAGs and reference genomes for the different taxonomic levels of archaea and bacteria was carried out, starting with the file comprising the multiple alignment of ubiquitous and single-copy proteins delivered by the classify_wf module of GTDB-Tk, which was used as input for RAxML v8.2.12 [43], using the PROTGAMMAAUTO model and 1000 bootstrap iterations. The generated trees were visualized using iTOL v6.7.3 [44].

To deepen the taxonomic characterization performed by GTDB-Tk, the ANI and AAI values between selected MAGs and their closest reference genome were calculated using FastANI v1.32 [45] and FastAAI v0.1.20 (https://github.com/cruizperez/FastAAI, accessed on 4 May 2023), respectively. In addition, the search for the closest published genome to the selected MAGs was performed using the TypeMat, NCBI Prok, and MAGs databases of the MiGA v1.1.2.2 tool [46].

## 3. Results

### 3.1. Microbial Diversity in Salar de Ascotán’s Water, Sediment, and Soil, as Revealed by 16S rRNA Metabarcoding

To investigate the microbial communities inhabiting Salar de Ascotán, we collected water, sediment, and soil samples from the stream six area (Table 1) and conducted 16S rRNA metabarcoding analyses, with the universal primers Bakt_341F and Bakt_805R directed to the V3-V4 region. More than 39 bacterial phyla were identified, anticipating a high microbial diversity, with a clear predominance of *Pseudomonadota*, especially in water; this corresponded to more than 75% of the taxa (Figure 1A). This situation was contrasted by a markedly higher abundance of *Bacteroidota*, *Spirochaetota*, *Chloroflexota*, and *Bacillota* in sediment and soil. Additionally, *Cyanobacteriota* and *Verrucomicrobiota* were abundant in sediment and water but less represented in soil, while *Deinococcota* was especially abundant in soil.

Each sample showed a different composition at the genus level, with a limited number of shared bacterial genera (Figure 1B). In soil, 99 genera were identified (≥10 ASV counts) (Appendix A), with *Halanaerobium* (*Bacillota*), *Aliifodinibius* (*Bacteroidota*), *Truepera* (*Deinococcota*), and *Thiohalophilus* (*Pseudomonadota*) predominating. In sediment, 156 genera were identified; the most abundant were *Thiobacillus* and *Leptothrix* (*Pseudomonadota*), followed by *Ignavibacterium* (*Bacteroidota*) and *Hydrogenophaga* (*Pseudomonadota*). A smaller number of genera (78) were found in water, with *Rhodoferax*, *Pseudomonas*, *Acinetobacter*, and *Rheinheimera* (all *Pseudomonadota*) predominating.

Regarding Archaea, a limited number of reads from this domain were found in the three samples using the Bakt primers, although they were considerably more in soil. Thus, we performed 16S rRNA metabarcoding of the soil sample using the primers Arc_787F and Arc_1059R against the V5–V7 region, shown to target archaeal taxa preferentially [47,48]. In total, 98.5% of the archaeal ASV counts corresponded to the phylum *Euryarchaeota*, 0.95% to *Nanoarchaeota*, and 0.58% to Candidatus *Asgardarchaeota*. Thirty genera were identified, with *Halorubrum*, *Halobellus*, *Halomicroarcula*, *Haloplanus*, and *Halolamina* (all *Euryarchaeota*) predominating (Figure 1C).

In terms of alpha diversity, the Shannon index ranged from 5.09 (water) to 6.54 (sediment), while the Simpson index was close to the unit in all the samples (Figure 1D). This result indicates an overall high diversity that is slightly lower in water, and the dominance of a relatively small number of genera. According to the relative abundance measurements, these taxa were different for each sample, suggesting that variations in the local environment conditions would select a different set of microorganisms, outstanding the presence of Archaea in soil.

### 3.2. Metagenomic Analyses and Captured Sequence Diversity

Next, we aimed to investigate the genomic diversity of this environment and reconstruct genomes of the predominant microbes to learn more about their taxonomy and predicted functional properties. We combined Illumina and Nanopore shotgun sequencing of the sediment, soil, and water samples, yielding 9.75–12.62 Gbp of Illumina data and 6.71–9.77 Gbp of Nanopore data (average read length: ~6.7 Kbp) (Appendix A). Using the Nonpareil algorithm, we obtained an annotation-independent measurement of microbial DNA sequence diversity and estimated the coverage reached for each metagenome with the sequencing effort made. The Nonpareil diversity index Nd was similar in the three samples but slightly higher in sediment, in agreement with the 16S-rRNA-based diversity calculations (Figure 1D). The observed values (21.0–22.3) are in the range of marine environments and some low-complexity soils [26].

The estimated coverage of the sediment metagenome was the lowest (51%), followed by 61% in water and 67% in soil (Figure 1E). It is worth noting that this coverage was calculated only based on the Illumina data since Nonpareil is not suited for long reads [26]. However, the Illumina data were complemented with Nanopore data for the downstream assembly-based analyses, thus reaching a higher total coverage. Upon comparing the sequence composition of the three metagenomes based on Mash distances [24], a closer relationship was observed between sediment and water (Figure 1F). In addition, Microbecensus analysis indicated an average genome size of ~5 Mbp for these two samples, while it was considerably smaller for soil (3.6 Mbp), in agreement with a markedly higher presence of archaea in this sample.

In agreement with 16S rRNA metabarcoding, the Illumina and Nanopore reads’ taxonomic classification followed an overall similar abundance pattern, predominating *Pseudomonadota* and *Bacteroidota* in the three metagenomes (Appendix A). Also, the phylum *Euryarchaeota* did not exceed 0.5% abundance in the sediment and water samples, while in the soil, it corresponded to 12.6% of the Nanopore reads and 14.8% of the Illumina reads.

### 3.3. Reconstruction and Taxonomic Novelty of Bacterial and Archaeal Genomes from Salar de Ascotán

The Nanopore and Illumina metagenomic reads were combined to obtain hybrid assemblies of 292, 330, and 697 Mbp for sediment, water, and soil, respectively (Appendix A). Remarkably, the assemblies had very low fragmentation (N50: 40–135 kbp), especially for soil, with the largest contig showing an extension of 5.2 Mbp and several contigs ≥ 1 Mbp. After contigs binning and bins refinement, 107 MAGs of quality ≥ 50% (completeness − 5 × contamination [49]) were obtained: 59 from soil (ASO), 33 from water (AWA), and 21 from sediment (ASE). Classification according to the Genome Taxonomy Database indicated 101 bacterial and six archaeal MAGs.

Bacterial MAGs were distributed in 16 phyla, with *Pseudomonadota* (54), *Bacteroidota* (30), *Actinomycetota* (10), and *Desulfobacterota* (10) predominating (Figure 2, Appendix A). They comprised 1–135 contigs and a total genome size ranging from 0.7 Mbp (ASO70, p_*Patescibacteria*; f_CSBR16-193) to 8.7 Mbp (AWA13, p_ *Actinomycetota*; g_*Rhodococcus*). The reconstructed genomes include several uncultured, poorly known candidate phyla radiation bacterial taxa. In this regard, we found eight MAGs from *Patescibacteria*, encompassing seven different orders. Among them, predicted taxonomic novelty at the family level was observed for the MAGs ASO57 and ASO83 from the orders UBA1400 and UM-FILTER-42-10. A similar situation was observed for ASO50 (*Acidimicrobiales*, *Actinobacteriota*) and ASO37 from order UBA4486 (*Pseudomonadota*). Additionally, ASO91 belonged to the recently proposed Candidate Phylum *Krumholzibacteriota*, whose first representative was recovered from an anoxic sulfidic spring [50]. We also identified ASO67 and ASO92 from the Candidate Phylum *Marinisomatota*, previously found in deep-sea environments [51,52], and ASO49 from Candidate Phylum KSB1, recently found in sulfur-rich anoxic environments and hypersaline microbial mats [53,54].

The six archaeal MAGs showed a genome size between 1.8 and 2.5 Mbp distributed in 1 to 29 contigs (Appendix A). According to GTDB-Tk analysis, five MAGs belonged to the class *Halobacteria* (ASO16, ASO38, ASO44, ASO48, and ASO85), from which ASO38, ASO48, and ASO85 were classified into three known families of the *Halobacteriales* order. Conversely, ASO44 corresponded to a novel family of this order, while ASO16 corresponded to a novel family of the JAHENH01 order. Thus, we examined the taxonomy of these MAGs in greater depth. First, they were analyzed with the Microbial Genomes Atlas (MiGA) platform [46], using the MAGs, TypeMat, and Candidatus databases. Also, we downloaded all the NCBI reference genomes of the class *Halobacteria* (313) and used them for phylogenetic inference and average amino acid identity (AAI) calculations, along with the five *Halobacteria* MAGs recovered in this study. As a result, ASO38 was grouped with the *Haloarculaceae* family, showing an AAI of ~74% with its closest genomes (GCA_009663395.1 and GCA_009184545.1, both from the genus *Halosegnis*) (Figure 3). Accordingly, MiGA indicated that this MAG would belong to a novel species from *Halosegnis* (*p*-value: 0.0097). ASO48 was grouped with members of the *Nitrialbaceae* family and presented an AAI of 55.32% with its closest genome (GCA_023008545.1, *Natribaculum luteum*), corresponding to a novel *Natribaculum* species (*p*-value: 0.0041) and possibly a novel genus (*p*-value: 0.36) from the *Nitrialbaceae* family.

ASO85 was grouped with members of the *Halobacteriaceae* family (Figure 3) and presented an AAI of 63.39% with its closest NCBI reference genome (GCA_001886955.1, f_*Halobacteriaceae*; s_*Halodesulfurarchaeum formicicum*). According to MiGA, this MAG represented a novel species (*p*-value: 0.0061) and possibly a novel genus (*p*-value: 0.44) from *Halobacteriaceae*. Meanwhile, ASO16 and ASO44 formed deep isolated branches and presented low AAI values (46.24% and 52.01%, respectively) with their closest genome (GCA_023008545.1, f_*Nitrialbaceae*; s_*Nitribaculum luteum*). According to MiGA, ASO16 probably belongs to a novel family (*p*-value: 0.055) and even a novel order (*p*-value: 0.19) from the *Halobacteria* class. Additionally, ASO44 belonged to a novel species (*p*-value: 0.001) and possibly to a novel family (*p*-value: 0.2) of the class *Halobacteria* (*p*-value: 0.017). In agreement, our phylogenetic analysis clearly showed that ASO44 and ASO16 cluster outside *Nitrialbaceae*, supporting their taxonomic novelty at the family level.

Regarding ASO22, assembled in a single contig (2.5 Mbp), GTDB-Tk analysis suggested it would belong to a new genus of the family PWKY01 (c_*Thermoplasmata*; o_ PWKY01). To further evaluate this possibility, we downloaded all the reference and the fully assembled genomes of the class *Thermoplasmata* from the NCBI database (30), inferring their phylogenetic relationships and calculating the pairwise AAI values. The ASO22 closest GTDB genome (GCA_020343715, AAI = 42.93%) was an unclassified *Thermoplasmata* archaeon, and both genomes clustered apart from the PWKY01 representatives (Figure 4). Moreover, according to MiGA, the closest genome corresponded to Candidatus *Hydrothermarchaeum profundi* (GCA_002011125, AAI = 48.88%), and most likely belongs to a novel family (*p*-value: 0.0047) and probably a novel class (*p*-value: 0.059) of the Phylum Euryarchaeota.

The results above indicate that Salar de Ascotán hosts poorly explored bacterial and archaeal communities with divergent genome sequences compared to the microorganisms currently in the databases. They include representatives with potential taxonomic novelty at the family, genus, and species rank. We provided high-quality MAGs of these taxa, with very low fragmentation, which could be used for further studies regarding the microbiota of hypersaline environments and their genomic and functional features.

### 3.4. Predicted Metabolic Capabilities and Molecular Functions Encoded in the Salar de Ascotán’s MAGs

We aimed to explore the metabolic potential and molecular functions encoded in the Ascotán MAGs. For this, the proteins encoded in each MAG were functionally categorized and mapped to a set of reference pathways using DRAM [39]. Also, we categorized the MAGs proteome in hierarchical subsystem levels using SUPER-FOCUS [40]. Both tools offer different and highly complementary functional categories and thus allow a deeper and broader genome profiling. In agreement with the high taxonomic diversity among the MAGs, diverse functional profiles were observed for the different lineages recovered from this environment. Regarding bacteria, we observed a markedly abridged set of functions in *Patescibacteria* (Figure 5, Appendix A), in agreement with their reported reduced genome and limited metabolic capabilities due to a probable symbiotic or parasitic lifestyle [55,56].

Concerning carbohydrate metabolism, *Bacteroidota*, some *Actinomycetota*, and KSB1 MAGs showed the highest versatility in terms of substrate use, as revealed by the number of carbohydrate-active enzymes (CAZy) present (Figure 5A). According to the prediction, members of these taxa metabolize cellulose, arabinans, glucans, polyphenolics, and xyloglucans, among other carbon sources. Concerning methanogenesis and methanotrophy, ASE5 (p_*Desulfobacterota*) was the only MAG predicted capable of transforming methanol to methane and, along with ASO84 (p_*Chloroflexota*; o_ *Anaerolineales*), converting monoethyl amine to ammonia. Furthermore, ASO3 (p_*Pseudomonadota*; g_JAABTG01) and ASO37 (p_ *Pseudomonadota*; o_UBA4486) stand out for their predicted ability to produce methane from CO_2_. This evidence suggests an anaerobic lifestyle based on methanogenesis or ammonification for these bacteria.

Regarding other functions predicted for the Salar de Ascotán microbiota, *Desulfobacterota* MAGs (from soil and sediment) showed genes for aerobic ammonia oxidation, sulfate reduction, and thiosulfate oxidation. These functions would be highly relevant for this ecosystem, as sulfate-reducing bacteria were shown to be important anaerobic carbon remineralizers in aquatic sediments of varying salinity [57]. Moreover, sulfate reduction is a key process in the carbon cycle of salt marsh sediments and a critical first step in nitrogen removal from estuarine and coastal environments [58].

Finally, arsenate reduction potential was observed for several MAGs across different phyla, especially *Actinobacteriota* and *Pseudomonadota*. Also, mercury reduction potential was observed in several *Pseudomonadota* and one *Desulfobacterota* MAG. Additional functional categories can be found in the complete DRAM output shown in Appendix A.

SUPER-FOCUS profiling of the bacterial MAGs revealed that amino acids, protein biosynthesis, fatty acids, and central carbohydrate metabolism are among the top-represented categories across most lineages. A similar situation was observed with stress-related functions, including DNA repair, heat shock, oxidative stress, and resistance to antibiotics and toxic compounds (Appendix A). Conversely, we found an overall low content of virulence, disease, and defense factors in all the MAGs, while flagellar motility was more represented among *Pseudomonadota*. Of note, despite their reduced genome, SUPER-FOCUS revealed that *Patescibacteria* MAGs have a high abundance of genes encoding phage proteins, proteins involved in synthesizing capsular and extracellular polysaccharides, and cold and heat shock proteins.

Considering the high proportion of proteins involved in resistance to toxic compounds found among the MAGs, and given the high concentrations of arsenic and other metals reported in Salar de Ascotán, we performed a dedicated search for resistance genes, using the Antibacterial Biocide and Metal Resistance Genes database (BacMet) as reference. BLASTx analysis indicated that, in sum, the bacterial MAGs comprised more than 100 different gene families linked with resistance to copper, arsenic, mercury, cadmium, zinc, and tungsten, among other metals and toxic substances (Figure 5B). These families included the genes *ars*, *mer*, and *cop*, among other extensively studied metal resistance determinants [59,60,61]. All this evidence indicates that the Salar the Ascotán microbiota have evolved a rich repertoire of genes contributing to their ability to thrive in the harsh conditions found in this environment. These genes encode proteins in charge of alleviating the oxidative stress and DNA damage derived from high salinity and UV radiation, the cold and heat shock derived from high thermal oscillations, and the toxicity of arsenic and other harmful substances present.

The functional profiling of the archaeal MAGs was less informative than for bacteria, probably due to an underrepresentation of this domain in the DRAM and SUPER-FOCUS databases. Nonetheless, DRAM showed a contrasting profile when comparing ASO85 with the rest of the MAGs, as it was the only one with electron transport chain (ETC) complexes III and IV (high affinity), together with the I, II, and IV (low affinity) complexes present in most of the MAGs (Appendix A). Among other functions detected in part of the archaeal MAGs, ASO16, ASO44, and ASO48 had genes for nitrite reduction (to nitric oxide), ASO22 had genes for thiosulfate to sulfate oxidation, ASO85 had genes for thiosulfate to sulfite dismutation, and ASO38 had genes for mercury reduction. Additionally, SUPER-FOCUS showed for all the archaeal MAGs an overall high abundance of proteins involved in oxidative stress and resistance to toxic compounds (Appendix A). However, ASO22 differed from the others in having a higher proportion of CRISPR-CAS, capsular and extracellular polysaccharides, Fe-S clusters, and flagellar motility, although fewer genes linked to cofactors, vitamins, prosthetic groups, and pigments. Regarding genes for metal resistance, the search using the BacMet database produced only hits with low identity and coverage, probably due to a marked underrepresentation of archaeal sequences in this database.

## 4. Discussion

When studying microbial diversity at the genetic level, it is fundamental to consider that different processes have impacted it throughout evolution. These processes range from gradual mutational adaptive evolution and vertical inheritance to the horizontal transfer of genes and mobile genetic elements, introducing shorter timescale variability and abrupt phenotypic changes [62]. Something that characterizes the salt flats in the South American Altiplano is the environmental heterogeneity that acts as a selective factor, with marked variations in the availability of water, minerals, saline concentration, temperature, and UV radiation [1,63]. Together, these factors generate complex environmental conditions that fluctuate cyclically in the niche, affecting the structure of microbial communities [64,65]. It should be noted that thermal oscillations between day and night intensify the natural salinization process. This phenomenon, in turn, influences the composition of biotic communities, characterized mainly by the prominent presence of halophilic bacteria and archaea [66]. While previous studies focused on the structure of the Salar de Ascotán microbial communities, this work contributed with a set of high-quality MAGs to unveil the genomic features, encoded functional capabilities, and taxonomic novelty of the predominant bacterial and archaeal lineages inhabiting this model extreme environment.

According to our analyses, a predominance and higher diversity of bacteria than archaea was observed in sediment, water, and soil. In agreement with previous studies based on 16S rRNA metabarcoding, the phyla *Pseudomonadota*, *Bacillota*, *Bacteroidota*, and *Actinomycetota* stood out [6,63]. These phyla, along with *Chloroflexota*, are commonly found in salt flats, as they can thrive in high-salinity conditions [66]. Furthermore, phyla such as *Bacteroidota* and *Pseudomonadota* are prevalent in the microbial communities of high-altitude water bodies, and can adapt to low temperatures and high UV-B radiation. Other phyla highly represented were *Deinococcota* and *Verrucomicrobiota*, which also inhabit other altiplanic salt flats, as is the case of the Salar de Huasco [6].

Although a very different composition at the genus level was observed in the three biomes studied, several of the predominant genera identified have been found in other high-salinity environments. Among *Pseudomonadota*, we found *Halomonas*, *Rhodoferax*, *Marinobacter*, *and Thioalkalivibrio*. The genus *Halomonas* includes Gram-negative, halophilic, and halotolerant bacteria detected in habitats with varied pH, temperature, and salinity gradients. Moreover, *Halomonas* strains exhibiting remarkable UV radiation resistance have been isolated from hypersaline environments [67,68]. In addition, the genus *Marinobacter* comprises chemoheterotrophic and halophilic species isolated from saline soils, sands, hypersaline lakes, and oil wells [69], and was previously found in water samples obtained from Salar de Ascotán [63]. Additionally, *Rhodoferax* species were identified in water from moderate to high-salinity lakes in the Tibetan Plateau [70], and the genus *Thioalkalivibrio* was reported in alkaline, high-salinity lakes [71].

In addition, we found a high proportion of the *Bacteroidota* genera *Psychroflexus*, *Salinibacter*, and *Aliifodinibius*, which have attracted particular interest due to their role in the mineralization of organic matter in marine environments. Previous studies conducted in ecosystems of the Chilean Altiplano have documented the presence of the psychrophilic genus *Psychroflexus*, which was also previously found in Antarctica [72] and in environments with high salinity, including Salar de Ascotán, where it coexists with the genus *Salinibacter*. The last-mentioned genus has been detected in high-altitude lakes and is characterized by its preference for habitats with high salt concentrations [7,64,73]. Also, *Aliifodinibius* species have been isolated from different hypersaline environments and are considered moderately halophilic bacteria [74,75,76].

Concerning *Bacillota*, we found that *Halanaerobium* is the most abundant genera in Ascotán soil. Strains from *Halanaerobium* have been described as extremely halophilic, alkalitolerant, and strictly anaerobic, previously being found in hypersaline lakes, with biotechnological applications in water treatment [77,78].

Our results indicate that Ascotán soil hosts a remarkable archaeal community, with a clear predominance of *Euryarchaeota*. This phylum, one of the two main groups in the Archaea domain, comprises a physiological diversity that includes aerobes, anaerobes, chemoautotrophs, and heterotrophs. Within their phyla are extreme thermophiles, acidophiles, and thermoacidophiles [79]. Previous phylogenetic analyses of the archaeal community in Ascotán sediment samples have pointed to the predominance of this phylum [63]. In particular, we found the genus *Halorubrum* predominating in soil, which was reported in saline and hypersaline soils [66]. In Salar de Ascotán, this genus was detected in significant quantities in soil samples, showing a high tolerance to salinity [6]. Some related genera within this phylum, such as *Haloplanus*, *Haloarcula*, *Halobellus*, and *Natronomonas*, are found in high-salinity soil and water [80].

It has been suggested that archaea are an essential part of the microbiota of the gastrointestinal tract of vertebrates and play an important role in their health [81]. However, little is known about the diversity and composition of the archaeal community that lives in the intestines of birds, especially in wild specimens [82]. Given that there is a high prevalence of archaea in the soil of the Salar de Ascotán (relative abundance = 15%) and in similar environments, such as in the sediment of the Salar de Huasco (relative abundance = 5%) [83], part of the archaeome of resident and migratory birds is likely acquired from this type of environment enriched in archaea. Therefore, more research is required to investigate the microbiota of local fauna in search of a possible link with the environmental microbiome, which may be crucial for the survival of the organisms that live there. This information is highly relevant for future conservation efforts.

Until now, the unique microbial communities of Salar de Ascotán were mainly only known to be there, with scarce information regarding their molecular features, especially for uncultured taxa. Combining Nanopore and Illumina sequencing and a hybrid metagenome assembly strategy, we obtained high-quality MAGs composed of fewer and larger contigs. These valuable data gave us a detailed panorama of their encoded properties and can be used in future comparative genomic analyses. Furthermore, this MAG set could be leveraged for genome mining efforts, particularly when searching for enzymes or metabolites with biotechnological potential. In this regard, a recent study reported the characterization of two thermostable xylanases resistant to freeze-drying cycles from a halotolerant *Bacillus* sp. isolated from Salar de Ascotán, with proposed industrial applications [84]. Moreover, among our set of MAGs, we found several forming deep isolated branches within their respective lineages, with proposed taxonomic novelty at high taxonomic ranks. Hence, this information will contribute to expanding biodiversity databases with novel taxa and provide baseline knowledge regarding the microbiota of the underexplored extreme environments of the global South.

Using complementary functional profiling strategies, we obtained a detailed catalog of the MAGs’ encoded features, giving an ample view of the functions of the Salar de Ascotán microbiota and their metabolic peculiarities. Considering the high number of MAGs, covered phyla, and the numerous functional categories required to classify and quantify this genetic and functional diversity, this kind of analysis provides vast information that could be revisited and reinterpreted in the context of further studies focused on specific taxa or processes. Therefore, besides the specific functions and metabolic routes discussed in this work, this comprehensive functional profiling is a good starting point for studying the molecular adaptations of this microbiota to the extreme conditions found in this environment.

As expected, among the most represented functions in archaeal and bacterial MAGs were those related to stress response, including DNA repair, oxidative stress relief, and resistance to toxic substances. These functions correlate with the high UV radiation and salinity, the wide temperature variations, and the high concentration of toxic substances, including arsenic and other toxic metals.

Arsenic is a highly toxic metal present in elevated concentrations in Salar de Ascotán and other hypersaline environments in the Atacama Desert, where resistant bacteria and arsenic metabolizers have been found [4]. Microorganisms have developed several strategies to cope with arsenic toxicity. They can metabolize inorganic arsenic through oxidation, reduction, methylation or demethylation processes, including extracellular precipitation, chelation, intracellular sequestration, active extrusion, and biochemical transformation, to develop resistance or generate energy through arsenite oxidation or arsenate reduction. Although the arsenic metabolism of microbes in alkaline saline environments has been extensively investigated, less attention has been paid to lower pH saline environments, such as Salar de Ascotán [4,60].

Among the microbial genes previously linked to arsenic metabolism, *arsABCDH*, *acr3*, *arsM*, *aioA*, and *arrA* participate in the reduction, oxidation, and extrusion of this toxic element, which are fundamental in the mitigation of its toxicity [5,85,86]. The presence of specific microbial populations, such as those belonging to the phyla *Bacillota* and *Pseudomonadota*, has been observed in sediments with high arsenic content, suggesting their relevance in the biogeochemical cycling of this metal and their ability to tolerate it in high levels [4]. In this regard, the tolerance to high concentrations of both arsenite (AsIII) and arsenate (AsV) was demonstrated for an *Exiguobacterium* strain isolated from Salar de Huasco [86] in the Atacama Desert. In agreement, we found most of these genes present among the MAGs recovered from Salar de Ascotán. Thus, these reconstructed genomes representative of Ascotán’s microbiota can be used in future studies to investigate, in a comparative genomic context, the genes allowing these extremophiles to deal with the toxicity of arsenic and several other metals.

Overall, this study contributed to unveiling the genomic and functional features of the microbiota living in Salar de Ascotan, including several halophilic and halotolerant taxa previously found in other hypersaline environments and novel archaeal and bacterial taxa. Moreover, we showed that Salar de Ascotán soil is a niche particularly interesting to the study of archaeal diversity. Additionally, we found a transversally high proportion of genes involved in environmental stress alleviation and a remarkable variety of metal resistance genes. The set of high-quality MAGs reconstructed will be of use in future studies, deepening our understanding of these microbes and their processes.

## Figures and Tables

**Figure 1 microorganisms-11-02819-f001:**
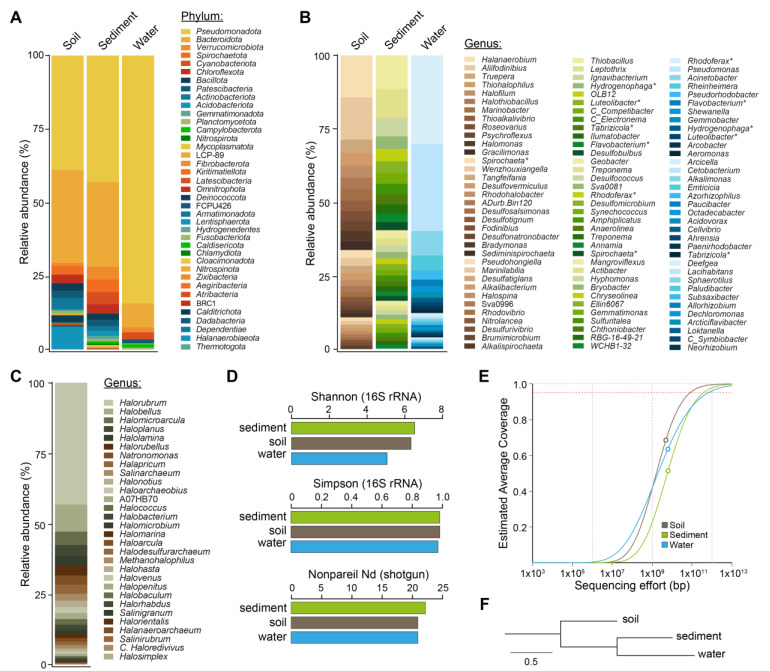
Microbial diversity in Salar de Ascotán soil, sediment, and water, as revealed by 16S rRNA metabarcoding and shotgun sequencing. Relative abundance of bacterial phyla (**A**) and genera (**B**) determined using the 16S rRNA Bakt_341F and Bakt_805R primers (V3–V4 region). Only the 30 most abundant genera in each sample were included for simplicity. The percentage of the total reads classified under these 30 genera corresponds to 81% in the case of soil, 58% in sediment, and 93% in water. * Genera present in two samples (mainly sediment and water). (**C**): Relative abundance of archaeal genera present in Ascotán’s soil, as determined using the Arc_787F and Arc_1059R primers (V5-V6 region). (**D**): Total (alpha) diversity according to the Shannon and Simpson indexes (calculated from the 16S rRNA gene metabarcoding data), and the Nonpareil Nd index from the analysis of shotgun metagenomic reads. (**E**): Nonpareil estimation of the metagenome coverage reached with the sequencing effort made for each sample. The circles correspond to the sequencing effort and the coverage reached for each sample, while the curves correspond to the fit according to the Nonpareil model. (**F**): Distance tree inferred from the MASH sequence composition analysis.

**Figure 2 microorganisms-11-02819-f002:**
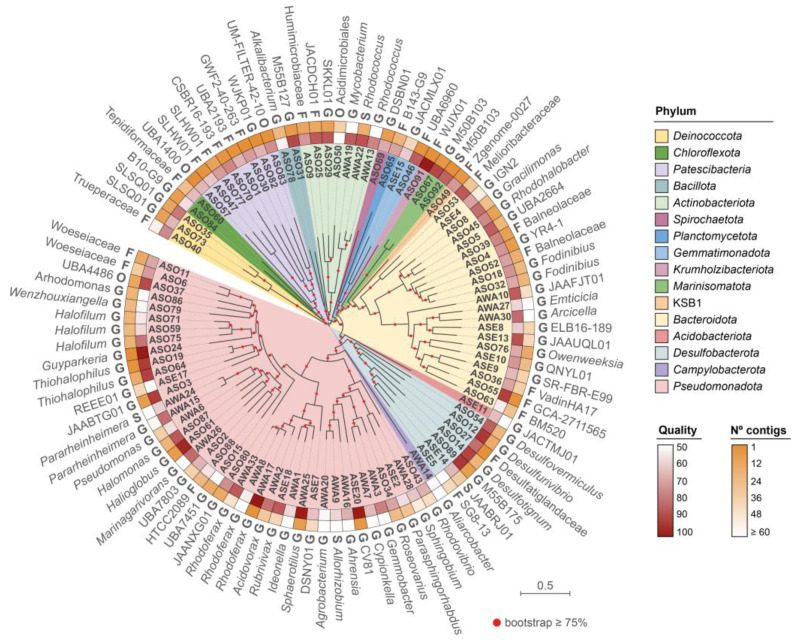
Phylogenetic relationships between the bacterial MAGs recovered from Salar de Ascotán’s soil, sediment, and water. The distance tree was inferred from the multiple sequence alignment of 120 bacterial marker proteins, as defined by the Genome Taxonomy Database classifier (GTDB-Tk). The colors of the branches correspond to different phyla, which were sorted in the legend according to the clockwise order they appear in the tree, starting from ASO40. The tracks (from inner to outer) denote the MAGs quality, number of contigs, and the lowest taxonomic rank at which it could be classified according to GTDB-Tk (F: family; G: genus; S: species), corresponding to the name shown in the outer track.

**Figure 3 microorganisms-11-02819-f003:**
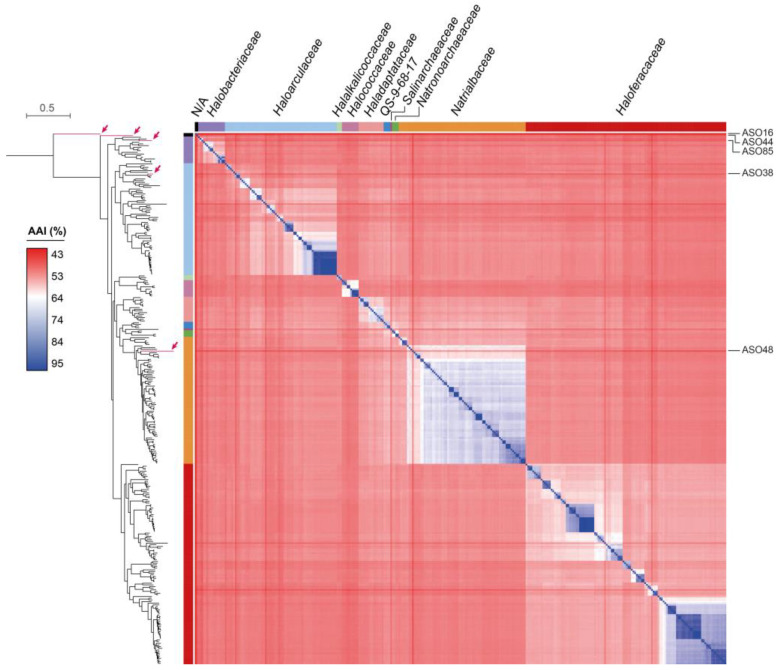
Phylogenetic relationships and AAI values among five *Halobacteria* MAGs recovered in this study (pointed with red arrows) and the 313 NCBI reference archaeal genomes from this class. The distance tree was inferred from the multiple sequence alignment of 53 archaeal marker proteins, as defined by the GTDB-Tk tool. The bootstrap values were omitted for more clarity. The different families are indicated with colors.

**Figure 4 microorganisms-11-02819-f004:**
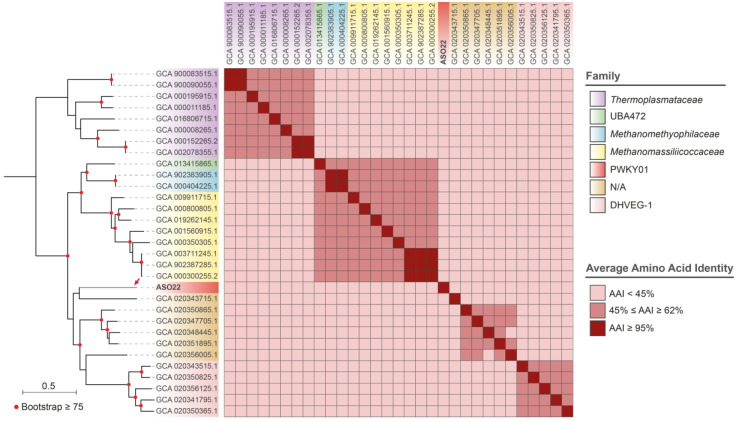
Phylogenetic relationships and AAI values among the MAG ASO22 (pointed with a red arrow) and the 30 NCBI reference archaeal genomes from the class *Thermoplasmata*. The distance tree was inferred from the multiple sequence alignment of 53 archaeal marker proteins, as defined by the GTDB-Tk tool. The bootstrap values correspond to 1000 iterations.

**Figure 5 microorganisms-11-02819-f005:**
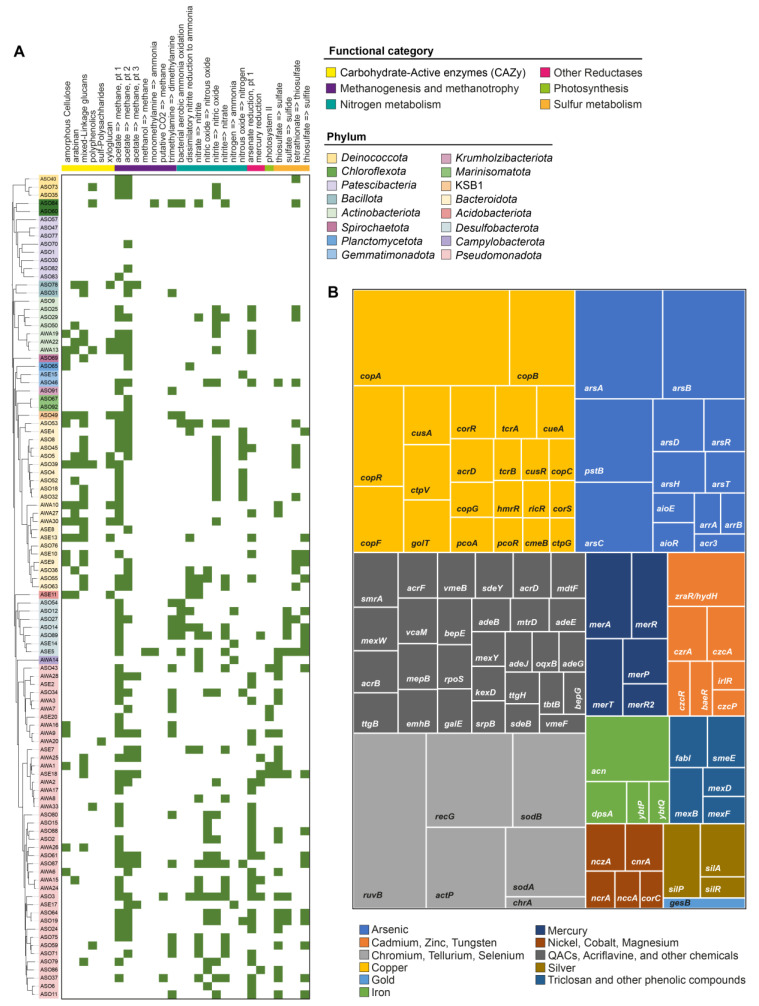
Predicted molecular functions encoded in the bacterial MAGs recovered from Salar de Ascotán. (**A**): Categorization of the metabolic pathways and molecular functions (selected categories) according to DRAM analysis. The heatmap denotes the presence/absence of the genes required for the corresponding function (panels to the right show). (**B**): Treemap showing the relative proportion of metal and biocide resistance genes found among the sum of the bacterial MAGs. The colors represent the element(s) and compound(s) to which the respective gene confers resistance.

**Table 1 microorganisms-11-02819-t001:** Samples and sampling site parameters.

Parameter	Water	Sediment	Soil
Location	S 21°29′52.7″ WO 68°15′25.3″	S 21°29′52.3″ WO 68°15′26.6″	S 21°29′53.7″ WO 68°15′26.9″’
Elevation (m.a.s.l ^1^)	3343	3350	3329
Environmental temperature (°C)	23.1	20.5	20.5
Water temperature (°C)	16	23.5	-
Pressure (HPa)	651.9	651.3	651.3
Humidity (%)	9.2	12.0	12.0
Soil or sediment temperature (°C)	-	22.0	17.5
pH	6.5	7.5	7.0
Water conductivity (mS)	3.93	-	-

^1^ Meters above sea level.

## Data Availability

The raw Illumina and Nanopore reads of the soil, sediment, and water metagenomes, along with MAGs described in this study, were deposited in the NCBI database under the BioProject accession PRJNA1023877.

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
