# Peer review of "Diversity, Taxonomic Novelty, and Encoded Functions of Salar de Ascotán Microbiota, as Revealed by Metagenome-Assembled Genomes"

_microorganisms, 2023, doi:10.3390/microorganisms11112819_

Round 1
Reviewer 1 Report
Comments and Suggestions for Authors
Overall the english is sufficient but can definitely be improved. The uses of certain words such as "However", "Also", etc need to be reviewed. Additionally, sometimes the structure of the results and discussion made it hard to follow.
Author Response
We thank the revision effort and the constructive criticism of the reviewer. Please see the attachment with our point-by-point response

Reviewer 2 Report
Comments and Suggestions for Authors
- Please avoid abbreviation in the abstract.
- Microbial classification names, such as phyla, orders, classes, etc. should be written in italic form. Please revise this issue in the whole manuscript.
For more information please check this article:
https://www.microbiologyresearch.org/content/journal/ijsem/10.1099/ijsem.0.005056
Names of the category “phylum” in the taxonomic hierarchy of bacteria were earlier not regulated in the [International Code of Nomenclature of Prokaryotes (ICNP)]. However, in February 2021, the members of the International Committee on Systematics of Prokaryotes (ICSP) have decided that naming of phyla must also be regulated in ICNP. The new names of 42 bacterial phyla were published in an article entitled: Valid publication of the names of forty-two phyla of prokaryotes by Ahron Oren and George M. Garrity. The following rules are important for naming of phyla:
1. In the names of phyla, -ota must be used as the ending.
2. Italics must be used for names of phyla in text.
3. A phylum name must be based on a genus, which constitutes the nomenclature type of the phylum in question.
- The abstract does not provide specific details why it is important to study microbial communities of the mentioned area. Moreover, why studying archaeal lineages is important in this area.
- The abstract mentions that Salar de Ascotán is an essential habitat for migratory birds and is critical for conservation and protection. However, it does not discuss how the findings of the study on microbial communities could inform conservation efforts or contribute to the understanding of ecosystem dynamics. It is highly recommended to add more details to make the abstract more attractive for the researchers and scientists.
- Please indicate the GPS (global positioning system) coordinates of Salar de Ascotán and the sampling areas.

Please check the attached file.
Author Response

(The authors gave the same response as above.)

Reviewer 3 Report
Comments and Suggestions for Authors
In this manuscript, the authors comprehensively characterizes the microbial communities of Salar de Ascotán, highlighting their diversity, uniqueness, and functional potential, outstanding novel archaeal and bacterial lineages. The author achieves this by analyzing the encoded functions in the recovered MAGs, finding a high proportion and variety of genes related to stress response and metal resistance. This study contributes to our understanding of the diversity and functional potential of microbiota in extreme environments and highlights the importance of studying these unique ecosystems. The manuscript is very well-written, however, there seems to be some concerns I have about this manuscript:
1. I think the novelty is not very obvious, maybe you could stress it in your manuscript.
2. Line 142, "2.316. S amplicon" should be corrected to "2.3. 16S amplicon". Additionally, in line 208, "Table S1)" should be revised to "(Table S1)".
3. If you try to avoid repeating information that has already been presented in the introduction and results section, the article may be more accessible for readers.
4. Why are there not at least three or more replicates for each sample type, such as water, sediment and soil?
Author Response

(The authors gave the same response as above.)

Round 2
Reviewer 1 Report
Comments and Suggestions for Authors
I appreciate the effort the authors made to address my concerns.